# Instability in NAD+ metabolism leads to impaired cardiac mitochondrial function and communication

Knut H Lauritzen[1]*, Maria Belland Olsen[1], Mohammed Shakil Ahmed[2], Kuan Yang[1], Johanne Egge Rinholm[3], Linda H Bergersen[4,5], Qin Ying Esbensen[6], Lars Jansen Sverkeli[7], Mathias Ziegler[7], Håvard Attramadal[2], Bente Halvorsen[1,8], Pål Aukrust[1,8,9], Arne Yndestad[1,8]

[1]Research Institute of Internal Medicine, Oslo University Hospital, Rikshospitalet and University of Oslo, Oslo, Norway; [2]Institute for Surgical Research, Oslo University Hospital and University of Oslo, Oslo, Norway; [3]Department of Microbiology, Oslo University Hospital, Oslo, Norway; [4]Department of Oral Biology, University of Oslo, Oslo, Norway; [5]Department of Neuroscience and Pharmacology, Center for Healthy Aging, University of Copenhagen, Copenhagen, Denmark; [6]Department of Clinical Molecular Biology, University of Oslo and Akershus University Hospital, Nordbyhagen, Norway; [7]Department of Biomedicine, University of Bergen, Bergen, Norway; [8]Institute of Clinical Medicine, University of Oslo, Faculty of Medicine, Oslo, Norway; [9]Section of Clinical Immunology and Infectious Diseases, Oslo University Hospital Rikshospitalet, Oslo, Norway

*For correspondence:
Knut.Huso.Lauritzen@rr-research.no

Competing interests: The authors declare that no competing interests exist.

**Abstract** Poly(ADP-ribose) polymerase (PARP) enzymes initiate (mt)DNA repair mechanisms and use nicotinamide adenine dinucleotide (NAD+) as energy source. Prolonged PARP activity can drain cellular NAD+ reserves, leading to de-regulation of important molecular processes. Here, we provide evidence of a pathophysiological mechanism that connects mtDNA damage to cardiac dysfunction via reduced NAD+ levels and loss of mitochondrial function and communication. Using a transgenic model, we demonstrate that high levels of mice cardiomyocyte mtDNA damage cause a reduction in NAD+ levels due to extreme DNA repair activity, causing impaired activation of NAD+-dependent SIRT3. In addition, we show that myocardial mtDNA damage in combination with high dosages of nicotinamideriboside (NR) causes an inhibition of sirtuin activity due to accumulation of nicotinamide (NAM), in addition to irregular cardiac mitochondrial morphology. Consequently, high doses of NR should be used with caution, especially when cardiomyopathic symptoms are caused by mitochondrial dysfunction and instability of mtDNA.

## Introduction

Myocardial dysfunction ultimately leading to heart failure (HF) is an increasing health concern worldwide. Although HF treatment has improved during the past few decades, the mortality and morbidity of this disorder is still high, suggesting that important pathogenic mechanisms are not fully modified by the current treatment modalities (*Marzetti et al., 2013*). Mitochondria are critical for the high energy demand of the heart, and defects in energy metabolism in cardiac mitochondria are seen in various forms of myocardial dysfunction (*Ren et al., 2010*). In addition, mitochondria are an important source of cellular production of reactive oxygen species (ROS). Whereas ROS are involved in physiological signaling cascades regulating various cellular and organ functions, enhanced production may induce oxidative DNA damage that could promote development of cardiomyopathy and

HF (*Fleming et al., 2017*). In fact, dysfunction of cardiomyocyte mitochondria is increasingly believed to be an important feature of progression of various forms of cardiac disease, and mitochondrial morphological disorganization and dysfunction have been found in HF patients (*Unno et al., 2009*). Mitochondria contain their own genome (mtDNA), and even though it is uncertain how instability of mtDNA affects mitochondrial function during development of myocardial dysfunction, it is vital that mtDNA remains relatively damage-free for proper mitochondrial function as shown by the evolution of mtDNA specific repair pathways (*Gredilla et al., 2010*).

DNA damage in the form of apurinic/apyrimidinic (AP) sites increases the risk of single-strand DNA breaks and is predominantly recognized by poly(ADP-ribose) polymerase (PARP) enzymes that again initiate repair mechanisms. PARP activity is dependent on nicotinamide adenine dinucleotide (NAD)$^+$ as a substrate and localizes to the mitochondria as well as the nucleus (*Cheng et al., 2013*; *Rossi et al., 2009*). High PARP activity has been speculated to drain NAD$^+$ reserves (*Karamanlidis et al., 2013*). NAD$^+$ is found in all living cells, and serves as a crucial coenzyme for enzymes that fuel reduction-oxidation reactions, carrying electrons from one metabolic intermediate to another, and as a substrate for enzymes such as PARPs and sirtuins (*Verdin, 2015*). SIRT1-5 are deacetylases that are central in the cellular defense against DNA damage and oxidative stress, by increasing antioxidant pathways and by facilitating DNA damage repair. Accordingly, sirtuins are shown to promote longevity and can mitigate many diseases related to aging and cardiovascular disease such as HF. As a consequence, depletion of the NAD$^+$ pool would cause cellular harm due to loss of sirtuin activity and a dysregulation of a number of protective pathways (*Karamanlidis et al., 2013*). SIRT3 resides in mitochondria where it regulates a number of mitochondrial proteins (*Cheng et al., 2013*), and loss of its activity due to suboptimal NAD$^+$ levels leads to de-regulation of important and varied molecular processes, including antioxidant systems, mtDNA repair, and mitochondrial dynamics. Thus, whereas NAD$^+$ is an important co-factor for enzymes related to antioxidant defense and repair mechanisms, increased PARP activity may in itself deplete NAD$^+$ stores. Agents that could maintain NAD$^+$ levels could therefore be an attractive therapeutic approach in disorders where increased PARP activity and decreased SIRT activity drive impaired mitochondrial function and increased mtDNA damage and ultimately cellular and tissue failure (*Cheng et al., 2013*; *Karamanlidis et al., 2013*).

A recent study showed that supplementation with the NAD$^+$ precursor nicotinamide mononucleotide (NMN) can partially normalize NAD$^+$/NADH ratios, and thereby restore SIRT3 activity and consequently mitochondrial function in mice with HF (*Karamanlidis et al., 2013*). Also, studies using nicotinamide riboside (NR) as a means to boost NAD$^+$ levels have recently shown beneficial effects for several physiological functions, including cardiac function in mice with dilated cardiomyopathy (*Diguet et al., 2018*; *Zhang et al., 2016*). However, the therapeutic benefit and safety of these compounds are far from clear, and if and how NR supplementation improves cardiac function needs to be further elucidated.

In this study, we show a direct link between mtDNA damage to loss of mitochondrial function and communication, with cardiac hypertrophy as a consequence. We used a previously characterized transgenic mouse model (*Lauritzen et al., 2015*), where a mutated DNA repair enzyme (termed mutUNG1) under control of the Tet-on system creates high levels of AP sites specifically in the mtDNA of cardiomyocytes. We demonstrate that high levels of cardiomyocyte mtDNA damage cause a reduction in NAD$^+$ levels in heart tissue due to highly active DNA repair, and consequently mitochondrial dysfunction due to loss of activation of crucial proteins involved in mitochondrial homeostasis. In addition, we show that treatment with a high dose of NR as a tool to increase NAD$^+$ levels may inhibit rather than increase sirtuin activity due to accumulation of nicotinamide (NAM). Our findings suggest that NR might have disadvantageous effects on cardiomyocyte mitochondria, at least in high dosages.

## Results

### Elevated PARP activity depletes cardiac NAD$^+$ levels and increases mitochondrial protein acetylation

We utilized a mouse model where high levels of mtDNA damage in the form of AP sites are generated specifically in cardiomyocytes. These mice develop cardiac hypertrophy and die of HF ~8 weeks

after mtDNA damage initiation (*Lauritzen et al., 2015*). AP sites increase the risk of single-strand DNA breaks, which are recognized by PARP (*Caldecott, 2008*). PARP initiates DNA repair and uses NAD$^+$ as a substrate for its activity (*Cheng et al., 2013*). We hypothesized that strongly increased PARP activity would deplete NAD$^+$ levels, which again would impair the function of other crucial NAD$^+$-dependent proteins such as sirtuins. Indeed, we found increased PARP levels in isolated mitochondria from heart tissue from mutUNG1-expressing mice compared to wild-type (Wt) littermates (*Figure 1A–B*), accompanied by reduced total NAD$^+$ levels in cardiac tissue as assessed by HPLC (*Figure 1C*). Sirtuins are also dependent on NAD$^+$ and SIRT3 is the major mitochondrial isoform that controls the activity of a number of different proteins through deacetylation. Whereas we saw an increase in SIRT3 protein levels in mitochondrial extracts from mutUNG1 cardiac tissue (*Figure 1D–E*), the overall protein acetylation status was increased in mitochondrial extract from mutUNG1 (*Figure 1F–G*), indicating lower SIRT3 activity. This was also seen in the acetylation status of SOD2 (*Figure 1H–I*), which is dependent on SIRT3 activity. Impaired SOD2 function could contribute to weakened antioxidant defenses with enhanced oxidative DNA damage as a potential consequence. Importantly, increased damage will increase PARP activity, representing a potential vicious circle in these mice.

The cardiomyocyte mutUNG1 mouse model is characterized by severely disordered mitochondrial morphology (*Lauritzen et al., 2015*). OPA1 is an important protein involved in organization of the mitochondrial inner membrane and is also activated by SIRT3. By investigating the acetylation status of OPA1, we saw an increase in acetylation of immunprecipitated OPA1 in mutUNG1-expressing mice compared to Wt littermates in total cardiac tissue. This indicates that a loss of OPA1 activity and regulation play a part in the observed impaired mitochondrial morphology (*Figure 1J–K*).

## Elevation of NAD$^+$ levels through NR treatment does not mitigate cardiac hypertrophy in mutUNG1-expressing mice

To test if replenishment of NAD$^+$ levels could reverse the cardiac phenotype in mutUNG1-expressing mice, we included NR in their diet (referred to as chow-NR). NR enters the 'NAD salvage pathway' through nicotinamide riboside kinase 1 which metabolizes NR into NMN. NR can thereby be used as a means to boost NAD$^+$ levels (*Braidy et al., 2019*). The dose (400 mg/kg chow) was chosen from the literature and is considered a medium dose (*Gariani et al., 2016*; *Zhang et al., 2016*). To investigate if NR did boost NAD$^+$ levels in the animals, we first measured NAD$^+$ levels using HPLC in the myocardium. Even though we did reconfirm lower levels of NAD$^+$ in heart tissue in mutUNG1-expressing mice, we did not see any increase in cardiac NAD$^+$ levels with NR supplementation (*Figure 2A*). However, others have noted that it is difficult to achieve increase in NAD$^+$ levels in heart tissue, possibly due to very high metabolic turnover (*Trammell et al., 2016*). However, these authors found NR to increase hepatic NAD$^+$ levels. Indeed, we did detect an increase in NAD$^+$ levels in the livers of mutUNG1-expressing mice, suggesting that NR treatment does increase cellular NAD$^+$ in this model (*Figure 2B*), albeit in a potential tissue-specific manner. Nevertheless, NR supplementation did not lead to any significant changes in the cardiac phenotype of mutUNG1-expressing mice (*Figure 2C–K*). Physical and echocardiographic measurements show cardiac hypertrophy, with increased mass of both left and right ventricular size in mutUNG1-expressing mice compared to Wt littermates, but with no significant difference between mutUNG1-expressing mice fed chow with or without NR supplement. In Wt mice, NR supplementation in chow diet resulted in a tendency toward poorer cardiac function as compared with chow diet alone with reduced ejection fraction (*Figure 2I*, p=0.087) and fractional shortening (*Figure 2J*, p=0.080), but these differences did not reach statistical significance.

## NR treatment does not alleviate mitochondrial dysfunction in mutUNG1-expressing mice, but alters morphology in Wt mitochondria

Even though NR supplement failed to improve cardiac function in mutUNG1 mice, we wanted to investigate if NR could have an effect on the impaired mitochondrial morphology in these mice. For this, we performed electron microscopy on heart tissue sections from mutUNG1-expressing and Wt mice with or without NR treatment. In line with the lack of effect on SIRT3 activity, NR treatment did not seem to alleviate the phenotype of dysfunctional mitochondria in mutUNG1 mice, which display abnormal organelle shapes and severe damage of internal cristae structures of the mitochondria as

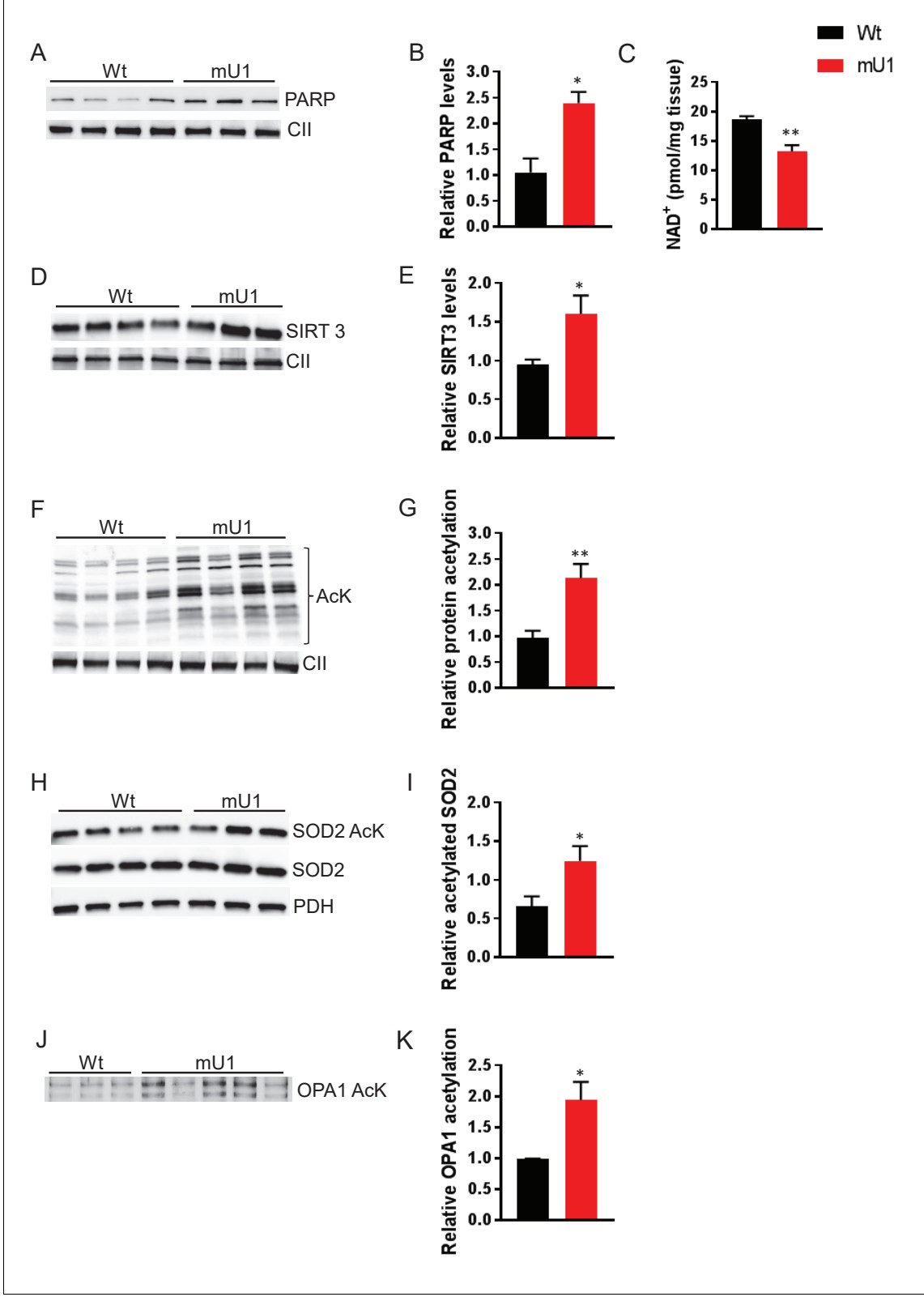

**Figure 1.** Elevated PARP activity depletes cardiac NAD$^+$ levels and reduces mitochondrial protein deacetylation. (A) Western blot showing PARP levels in cardiac mitochondrial extract from mutUNG1-expressing mice and wild-type littermates. (B) Quantification of PARP levels in western blot. (C) NAD$^+$ levels from heart tissue from wild-type and mutUNG1-expressing mice measured with HPLC. (D) Western blot of SIRT3 levels in extract of mitochondria isolated from cardiac tissue from mutUNG1-expressing mice and wild-type littermates. (E) Quantification of SIRT3 levels in western blot. (F) Western
*Figure 1 continued on next page*

*Figure 1 continued*

blot of protein acetylation levels in cardiac mitochondrial extract from mutUNG1-expressing mice and wild-type littermates. (**G**) Quantification of protein acetylation levels in western blot. (**H**) Western blot of SOD2 and acetylated SOD2 levels in cardiac mitochondrial extract from mutUNG1-expressing mice and wild-type littermates. (**I**) Quantification of relative acetylated SOD2 protein levels in western blot. (**J**) Western blot acetylated protein levels of samples of immunoprecipitated OPA1 from total extract (200 µg protein) of cardiac tissue from mutUNG1-expressing mice and wild-type littermates. (**K**) Quantification of relative acetylated OPA1 protein levels in western blot. Data is presented as mean ± SE. *p<0.05, **p<0.01 vs. Wt chow. Abbreviations: Wt = Wild-type mice, mU1 = mutUNG1-expressing mice, PARP = poly(ADP-ribose) polymerase, CII = mitochondrial complex II, SIRT3 = sirtuin 3, AcK = acetylated lysine, SOD2 = superoxide dismutase 2, PDH = pyruvate dehydrogenase, OPA1 = optic atrophy 1, and NAD$^+$ = nicotinamide adenine dinucleotide. Raw data are presented in *Source data 1*.

shown earlier (*Lauritzen et al., 2015*; *Figure 3A*). Whereas heart mitochondria from mice without NR treatment are tethered together with visible contact points, mitochondria in mice fed chow-NR seemed to have a rounder shape and to be organized in a more solitary fashion with less obvious contact points with neighboring mitochondria (*Figure 3A–B*). *Figure 3B* illustrates cristae structure of the cardiomyocyte mitochondria. Recent morphological studies have shown that adjacent mitochondria can interact through specialized regulated inter-mitochondrial junction (IMJ) sites, where cristae membranes become organized into coordinated pairs across organelles (*Picard et al., 2015*). This was also seen in our Wt mice, and highlighted in panel (ii) in *Figure 3B*, where there is a clear alignment of cristae structures of three neighboring mitochondria with visible contact points between mitochondria (indicated with red arrowheads). This, however, was not as prominent in Wt animals fed chow-NR, where the mitochondria have a rounder shape without tethering of neighboring organelles. The IMJs seem to be lost to a large extent, resulting in failing of cristae structures to align between neighboring mitochondria. As observed earlier (*Lauritzen et al., 2015*), the cristae structure in mutUNG1-expressing mice is heavily unorganized and damaged, and NR treatment did not alleviate this impairment (*Figure 3B*).

The electron microscope (EM) images were evaluated by a blinded scoring of cristae organization and mitochondrial shape. Internal cristae structure was found to be less organized and inter-organelle alignment was found to a lesser degree in mutUNG1-expressing mice compared to Wt animals, and not affected by NR supplement in the diet (*Figure 3C–D*). Mitochondria were also scored according to shape, and Wt animals fed NR had significantly more circular mitochondria than mice on a standard diet (*Figure 3E*). By measuring the length of electron-dense contact points (here defined as IMJ) and normalizing to total mitochondrial area, we also saw significantly lower IMJ length in all groups compared to Wt fed standard chow (*Figure 3F*). This was particularly surprising regarding Wt mice on an NR supplement, and we speculate that lower levels of IMJ are caused by the change of mitochondrial shape with a higher degree of circularity that leads to a loss of organelle tethering and thereby contact points (*Figure 3E*).

## Proteom analyses support structural effects of NR treatment on mitochondria

Mitofusin 2 (MFN2), dynamin-related protein (DRP1), and peroxisome proliferator-activator receptor gamma coactivator 1-alpha (PGC-1a) are proteins typically involved in mitochondrial morphogenesis and biogenesis. We quantified their proteins' levels in mutUNG1-expressing animals and Wt littermates (with and without chow-NR) in order to see if changes in mitochondrial structure could be related to changes in their expression levels. However, no significant changes could be detected (*Figure 4—figure supplement 1A–F*) and we continued with a broader approach performing MS-LC proteome analysis. Proteomic analysis of cardiac mitochondrial extract supported structural effects of NR treatment (*Figure 4A–B*). We calculated differentially expressed proteins (DEPs) for both mutUNG1-expressing and Wt mice given NR supplements vs. no supplement, and the DEPs were included in gene enrichment analyses (*Figure 4A*). For both Wt and mUNG1-expressing mice, the top two significantly enriched terms were *the citric acid (TCA) cycle and respiratory electron transport* (Reactome) followed by *mitochondrion organization* (gene ontology [GO]) (top five listed in *Figure 4A*). These two terms include overlapping proteins as illustrated in concomitant heatmaps, indicating crosstalk between these processes (*Figure 4A*). *Figure 4B* shows a simplified illustration (modified from *Pfanner et al., 2019*) including important regulators of the cristae structure, highlighting selected DEPs. Analysis is based on data presented in *Source data 1*.

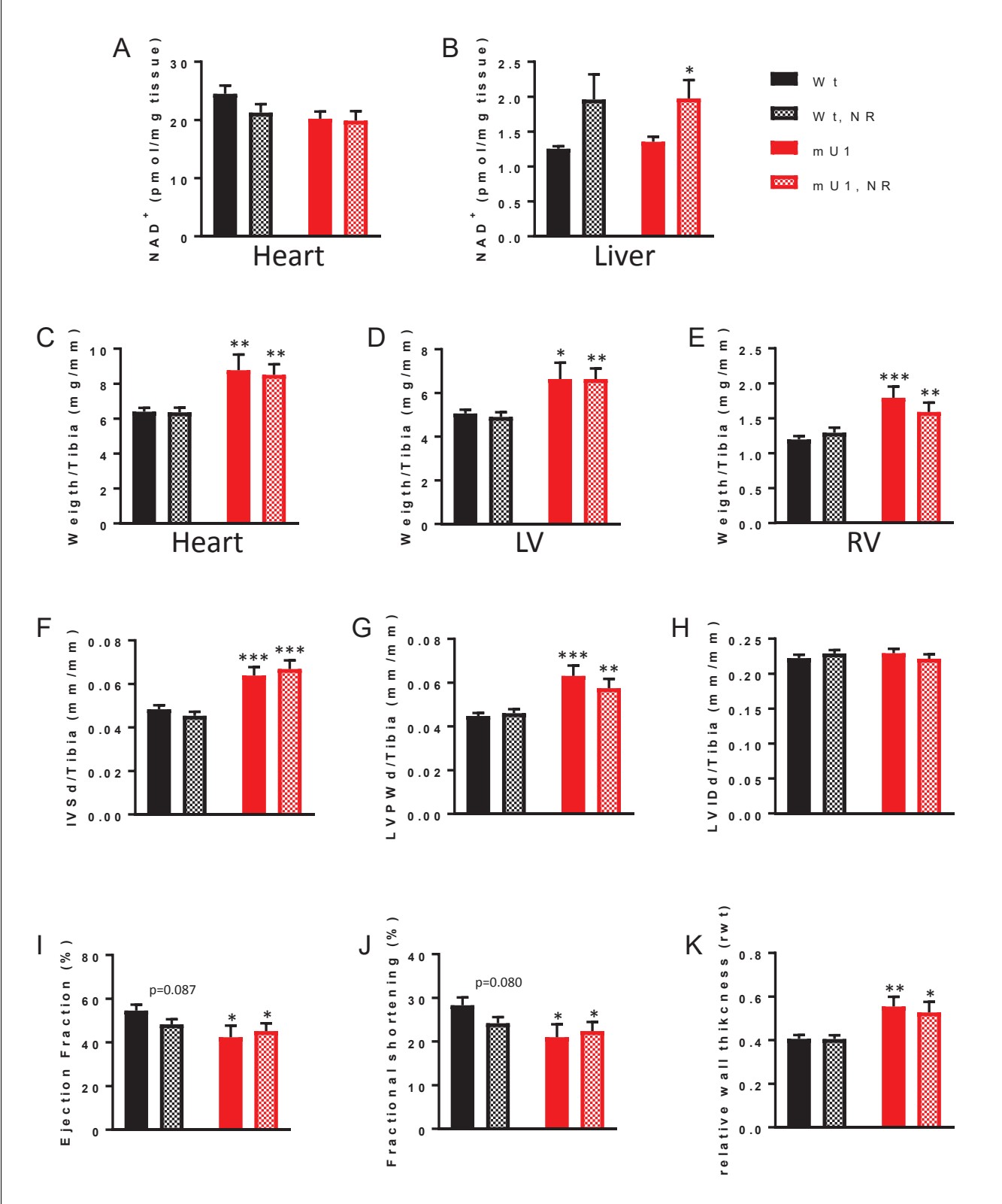

**Figure 2.** Elevation of nicotinamide adenine dinucleotide (NAD+) levels through nicotinamide riboside (NR) treatment does not mitigate cardiac hypertrophy in mutUNG1-expressing mice. NAD+ levels in heart tissue (**A**) and liver tissue (**B**) from wild-type and mutUNG1-expressing mice fed chow with or without NR, measured by HPLC. Weight of (**C**) heart, (**D**) left ventricle, and (**E**) right ventricle of wild-type and mutUNG1-expressing mice fed chow with or without NR. Echocardiographic measurement of (**F**) interventricular septum thickness at end-diastole, (**G**) left ventricular posterior wall

*Figure 2 continued on next page*

*Figure 2 continued*

thickness, (H) left ventricular internal dimension at end-diastole, (I) ejection fraction, (J) fractional shortening, and (J) relative wall thickness at end-diastole in wild-type and mutUNG1-expressing mice fed chow with or without NR. (C-H) Normalized against tibia length. Data is presented as mean ± SE. *p<0.05, **p<0.01, ***p<0.001 vs. Wt chow (for C-J, N = Wt; 15, Wt-NR; 17, mU1; 8, mU1-NR; 11). Raw data are presented in *Source data 2*.

We were also interested if NR treatment might alleviate the observed reduction in mtDNA copy number seen in mutUNG1-expressing mice (*Lauritzen et al., 2015*) but this was not the case (*Figure 4—figure supplement 2*). Additionally, earlier studies have shown a reduction in mitochondrial respiration in mutUNG1-expressing mice (*Lauritzen et al., 2015*), but analysis by high-resolution

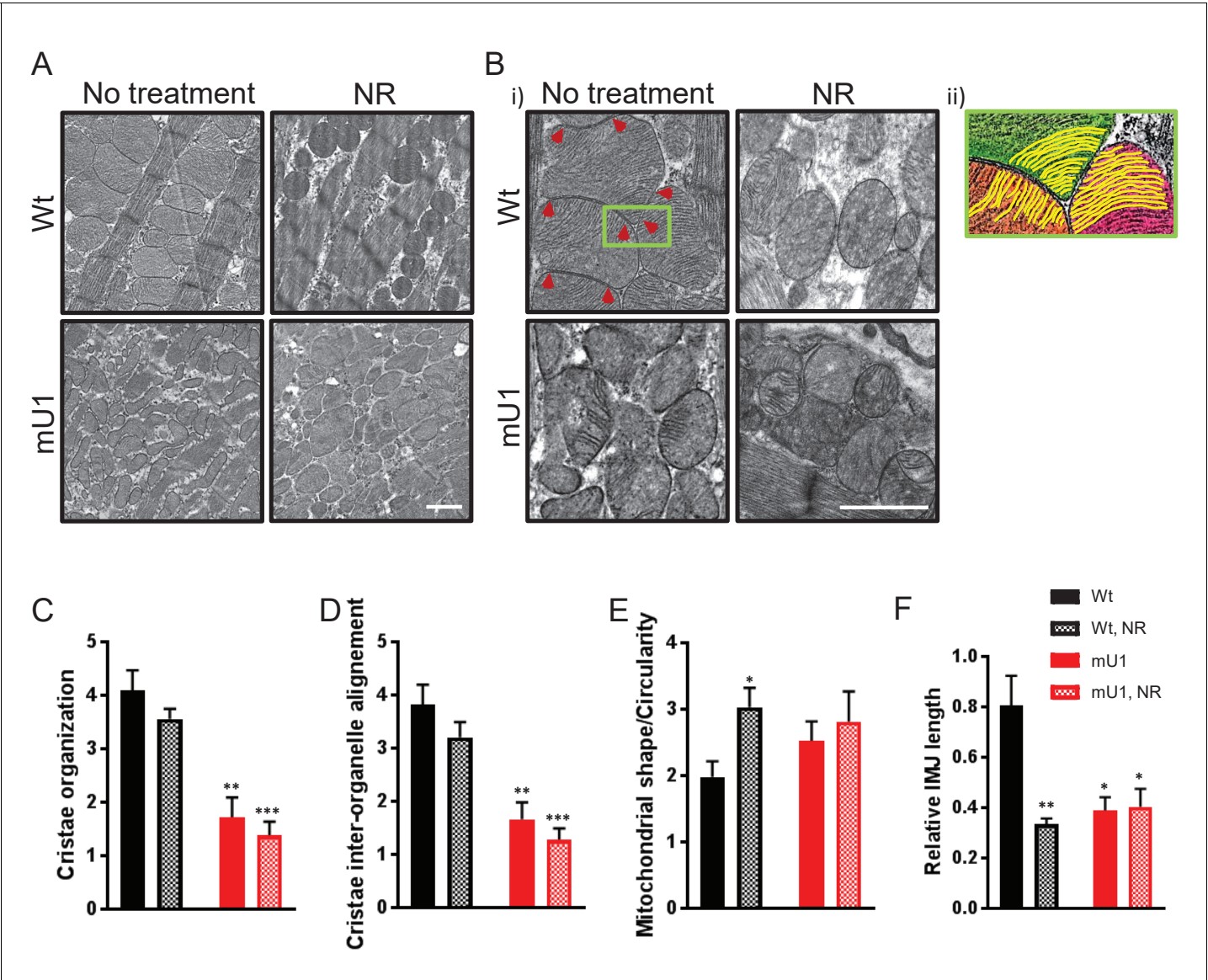

**Figure 3.** Nicotinamde riboside (NR) treatment does not alleviate mitochondrial dysfunction in mutUNG1-expressing mice, but does alter mitochondrial morphology in wild-type mitochondria. (A and B) Electron microscope images of wild-type and mutUNG1-expressing mice fed chow with and without NR. (B, ii) Detail of panel from (B) with an Illustration of aligned cristae (yellow) in three neighboring mitochondria (orange, green, and pink) in wild-type cardiac tissue. Electron-dense inter-mitochondrial junctions (IMJs) labeled with red arrowheads. The images are representative of five mice of each genotype and treatment. Scalebar = 1 μM. Analysis by scoring of (C) cristae organization, (D) cristae inter-organelle alignment, and (E) mitochondrial shape/circularity. (F) Quantification of relative IMJ length. Raw data are presented in *Source data 2*.

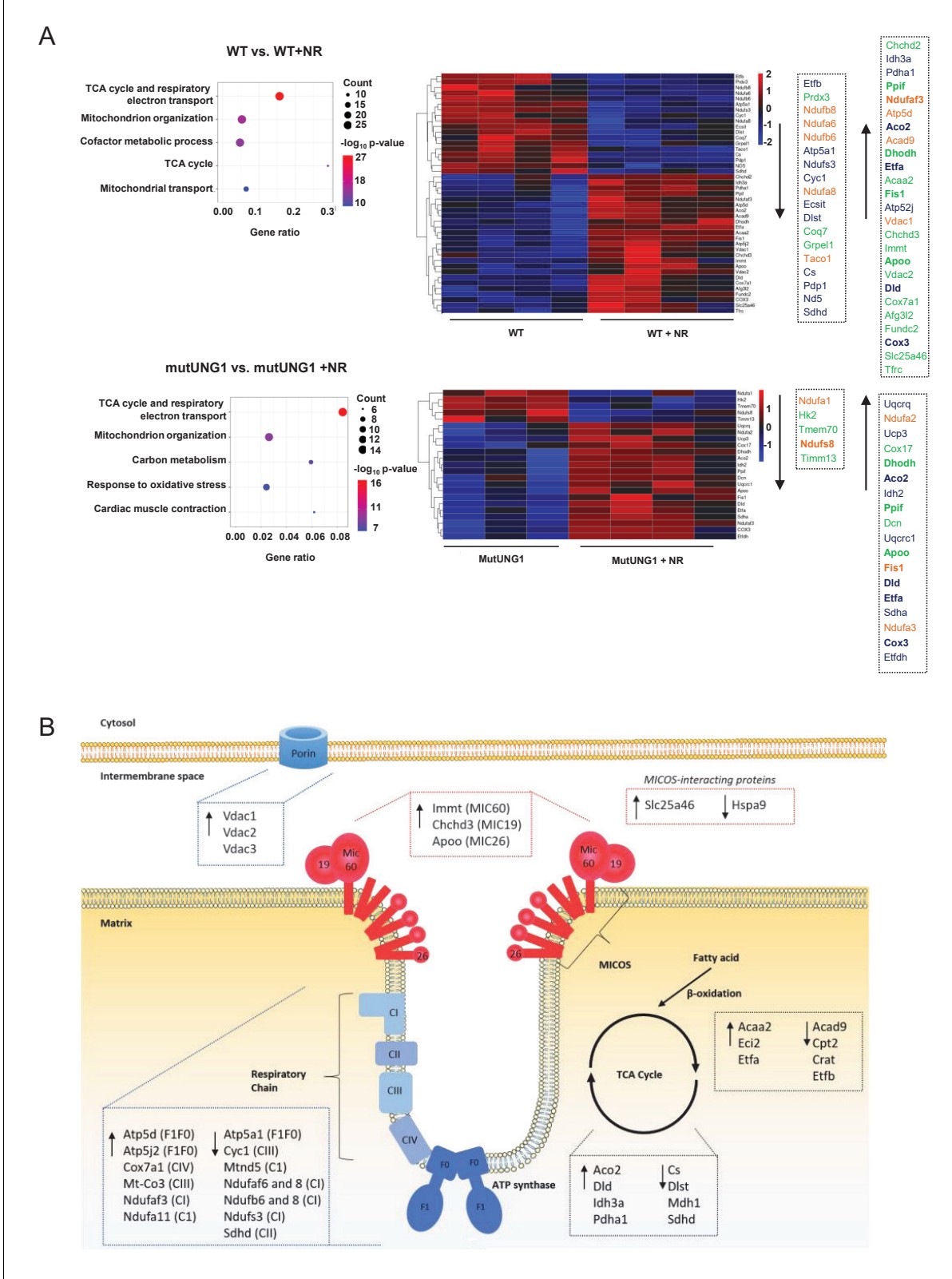

**Figure 4.** Proteom analyses support structural effects of nicotinamde riboside (NR) treatment on mitochondria. (**A**) Proteomic analysis of cardiac mitochondrial extract evaluating the effect of NR supplement in Wt and mutUNG1 mice. Shown are top five members by gene enrichment analysis. The differently regulated proteins (DEPs) included in the gene ontology terms *the citric acid* (*TCA*) *cycle and respiratory electron transport* (blue) and *mitochondrial organization* (green) are shown in the heatmap. Differentially expressed genes (DEGs) included in both pathways are marked in orange. *Figure 4 continued on next page*

*Figure 4 continued*

(B) A simplified illustration modified from *Pfanner et al., 2019* including important regulators of the cristae structure. The selection of DEPs is based on the gene enrichment analysis. Analysis is based on data presented in *Source data 1*.

The online version of this article includes the following figure supplement(s) for figure 4:

**Figure supplement 1.** Protein levels of mitofusin 2 (MFN2), dynamin-related protein (DRP1), and peroxisome proliferator-activator receptor gamma coactivator 1-alpha (PGC-1a) in mutUNG1-expressing mice and wild-type littermates.

**Figure supplement 2.** mtDNA copy numbers.

**Figure supplement 3.** Mitochondrial respiration.

respirometry after NR supplementation did not show increased respiration up to Wt levels (*Figure 4—figure supplement 3*).

Finally, the proteomic analysis was also used to verify the purity of the mitochondrial extract. From the 812 proteins, 28 were not linked to a GO term and excluded. From the 784 proteins left, 40 proteins could from their GO terms be connected to nucleus. Among these 40 proteins, 23 proteins did also have a GO term linking them to mitochondria, leaving 17 proteins connected to nucleus. Most of these could be further linked to the nuclear envelope. As mitochondria interact with other organelles, one could expect some contamination of non-mitochondria proteins in these extracts. However, as the level of nuclear proteins was very low (2.2%), and mainly linked to the nuclear membrane, we considered the purity of this extract to be satisfactory. Nuclear proteins detected in this analysis are shown in *Supplementary file 1*.

## NR treatment does not increase mitochondrial protein deacetylation in cardiac tissue, but causes accumulation of NM

We next examined the effect of NR supplementation on the acetylation status of mitochondrial proteins as an indicator of SIRT3 activity in cardiac tissue from mutUNG1-expressing mice. Whereas the acetylation was high in mutUNG1-expressing mice compared to Wt littermates, NR supplementation had no effect on the acetylation status (*Figure 5A–B*). To investigate if the NR dose was too low to have any effect on SIRT3 activity, we fed the animals with a higher NR dose (1000 mg/chow). This gave a small, non-significant increase in $NAD^+$ levels (*Figure 5—figure supplement 1*). To our surprise, a high dose of NR caused an even higher overall mitochondrial protein acetylation than the lower dose, at least in heart tissue of the mutUNG1-expressing mice (*Figure 5C–D*). However, this was not seen in liver tissue (*Figure 5—figure supplement 2A–B*). During deacetylation, sirtuins release NAM from their substrate $NAD^+$, and it has been shown that this can inhibit sirtuin activity as a negative feedback mechanism (*Avalos et al., 2005*). We therefore next measured NAM levels using HPLC and notably, there was a significant increase in NAM levels in cardiac tissue of mutUNG1-expressing mice, but not in Wt littermates fed chow-NR (*Figure 5E*). In liver tissue, NR-chow induced an increase in NAM levels in both genotypes, illustrating the complexity of $NAD^+$ metabolism and tissue differences in turnover (*Figure 5F*). No regulation of cardiac gene expression was found for NAMPT and NMAT1-3 (*Figure 5—figure supplement 3A–D*), indicating that accumulation of NAM is not caused by alterations in gene expression of components involved the salvage pathway in this context. Thus, it is possible that higher NR doses lead to an accumulation of NAM, and that this may result in inhibition rather than enhancement of SIRT3 activity, with an increase in the overall mitochondrial protein acetylation as a consequence.

Finally, we investigated the acetyl CoA levels in mutUNG1-expressing mice and Wt littermates on a chow-NR diet, but found no significant difference (*Figure 5—figure supplement 4*), indicating that acetylation of CoA is not implicated in the differences between the genotypes or in the effect of NR in these mice.

## Increasing doses of NR progressively inhibit deacetylation and PARP cleavage

Our findings so far suggest that increasing doses of NR could have non-beneficial effects on cardiac mitochondria. To further investigate this issue, we utilized a stably transfected cell line where mutUNG1 is expressed and under control of the Tet-on system (*Lauritzen et al., 2010*). As seen in heart tissue, the cells expressed PARP in mitochondria with a significantly higher level in mutUNG1-

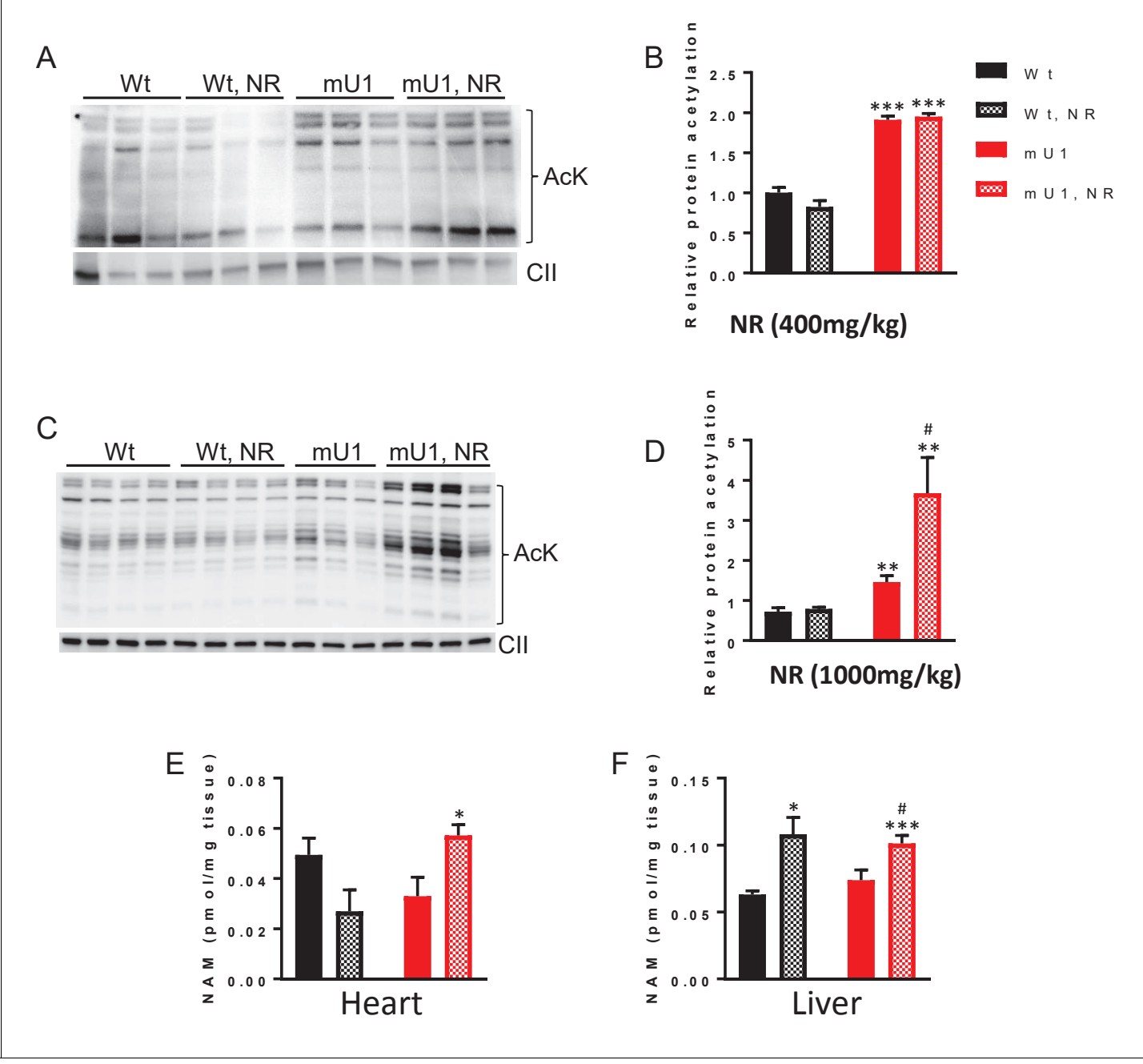

**Figure 5.** Nicotinamde riboside (NR) treatment does not counteract mitochondrial protein acetylation in cardiomyocytes but causes accumulation of nicotinamide. (**A**) Western blot of protein acetylation levels in cardiac mitochondrial extract from mutUNG1-expressing mice and wild-type littermates fed chow with or without medium dose of NR. (**B**) Quantification of protein acetylation levels in western blot. (**C**) Western blot of protein acetylation levels in cardiac mitochondrial extract from mutUNG1-expressing mice and wild-type littermates fed chow with or without high dose of NR. (**D**) Quantification of protein acetylation levels in western blot. Nicotinamide (NAM) levels in heart tissue (**E**) and liver tissue (**F**) from wild-type and mutUNG1-expressing mice fed chow with or without NR, measured with HPLC. Data is presented as mean ± SE. *$p<0.05$, **$p<0.01$ vs. Wt chow. #$p<0.05$ vs. mutUNG1 chow. Raw data are presented in *Source data 2*.

The online version of this article includes the following figure supplement(s) for figure 5:

**Figure supplement 1.** Nicotinamide adenine dinucleotide (NAD[+]) levels.

**Figure supplement 2.** Protein acetylation levels.

**Figure supplement 3.** Gene expression levels Nampt and Nmat 1–3.

**Figure supplement 4.** Acetyl CoA levels.

expressing cells (*Figure 6A–B*). Interestingly, the highest levels of PARP were in mutUNG1-expressing cells treated with NR (*Figure 6B*). By measuring PARylation as readout of PARP activity, we did see an increased activity in mutUNG1-expressing cells that could be silenced by the PARP inhibitor Olaparib (*Figure 6C–D*). PARP activity did not increase when supplementing NR (*Figure 6C–D*). As in cardiac mitochondria, we measured relative protein acetylation and observed that increasing doses of NR progressively inhibited deacetylation in cells, and to a higher degree in mutUNG1-expressing cells (*Figure 6E–F*). PARP inhibition reduced the acetylation levels to some degree, in line with these enzymes utilizing the same pool of $NAD^+$ (*Figure 6G–H*). SIRT3 silencing RNA (SiRNA) did reduce SIRT3 levels by more than 50% (*Figure 6I–J*). Accordingly, acetylation levels were increased following SIRT3 siRNA treatment, demonstrating SIRT3 does influence acetylation levels when high levels of NR are introduced (*Figure 6K–L*). All together, these results indicate stronger sirtuin inhibition with higher NR doses, in line with the observed inhibition in mouse cardiomyocyte mitochondria (*Figure 5C*).

## Discussion

Mitochondrial dysfunction has been suggested to play an important role in the progression of myocardial dysfunction, but the mechanisms for this 'cardiotoxic' effect are still not clear. Herein, we show that an increase in PARP activity in response to increased mtDNA damage depleted $NAD^+$ levels in mutUNG1 mice. These interactions resulted in decreased mitochondrial deacetylation, most likely due to impaired SIRT3 activity, which promoted further mitochondrial dysfunction, potentially representing a pathogenic loop in the progression of cardiac remodeling in these mice. While NR has been suggested to restore $NAD^+$ levels and thereby improve mitochondrial function and SIRT3 activity, we show that NR, particularly at high doses, had the opposite effects in cardiac tissue potentially secondary to enhanced NAM levels that would inhibit SIRT3 activity. Moreover, detailed studies of the mitochondria show that NR promoted disorganization of mitochondrial structures, involving impaired activation of OPA1, a key regulator of mitochondrial inner membrane structure. Our findings show the importance of PARP, $NAD^+$, and SIRT3 as well as their complex interactions in the development of mitochondrial dysfunction during deterioration of myocardial function. These data also suggest that the use of NR in rescuing these cardiac events should be reevaluated, in particular at higher dosages (*Figure 7*).

We wanted to investigate the correlation between loss of mtDNA integrity, mitochondrial dysfunction and cardiomyopathy, and utilized a previously characterized mouse model where mtDNA damage in the form of AP sites is induced specifically in cardiomyocytes in adult animals (*Lauritzen et al., 2015*). AP sites increase the risk of single-strand breaks, which are recognized by PARP that initiates $NAD^+$ consuming repair of this lesion (*Caldecott, 2008*; *Vida et al., 2017*). We therefore hypothesized that loss of $NAD^+$ could be a driving force in the fatal phenotype in this model. Interestingly, PARP (over)activation has been shown to contribute to HF induced by the anti-cancer anthracycline drug, doxorubicin (*Pacher et al., 2002*), and in vitro and in vivo studies with PARP inhibitors diminished this cytotoxic effect (*Ali et al., 2011*; *Damiani et al., 2018*). However, we found that NR supplementation with the intention to increase $NAD^+$ levels failed to improve the cardiac protein deacetylation status and phenotype in the mutUNG1 mice. In fact, a higher dosage of NR even worsened the cardiac deacetylation status in these mice. Importantly, NR caused a marked increase in NAM in both cardiac and liver tissue, and this compound has been shown to inhibit sirtuin activity (*Avalos et al., 2005*). It is therefore likely that NAM-mediated inhibition of sirtuin activity in the myocardium during NR supplementation results in impaired rather than improved cardiac protein deacetylation and function. An earlier study showed that even though the nuclear and cytoplasmic $NAD^+$ levels were reduced after genotoxic stress, the mitochondrial $NAD^+$ pool remained unchanged (*Yang et al., 2007*). We however speculate that specific mtDNA damage like AP sites (with concomitant PARP activation) will affect the mitochondrial $NAD^+$ levels as well.

Depletion of the cellular and mitochondrial $NAD^+$ pool can cause inhibition of important $NAD^+$-dependent proteins, especially the sirtuins. This seems to be the situation in our model, where there is a reduction of $NAD^+$ levels in cardiac tissue of mutUNG1 expressing mice and a loss of mitochondrial deacetylation. Even though there seems to be elevated protein levels of SIRT3 in these mice, this might be a cellular compensatory strategy due to reduced deacetylation activity, even though the regulation of mitochondrial proteins through acetylation seems to be complex (*Fisher-*

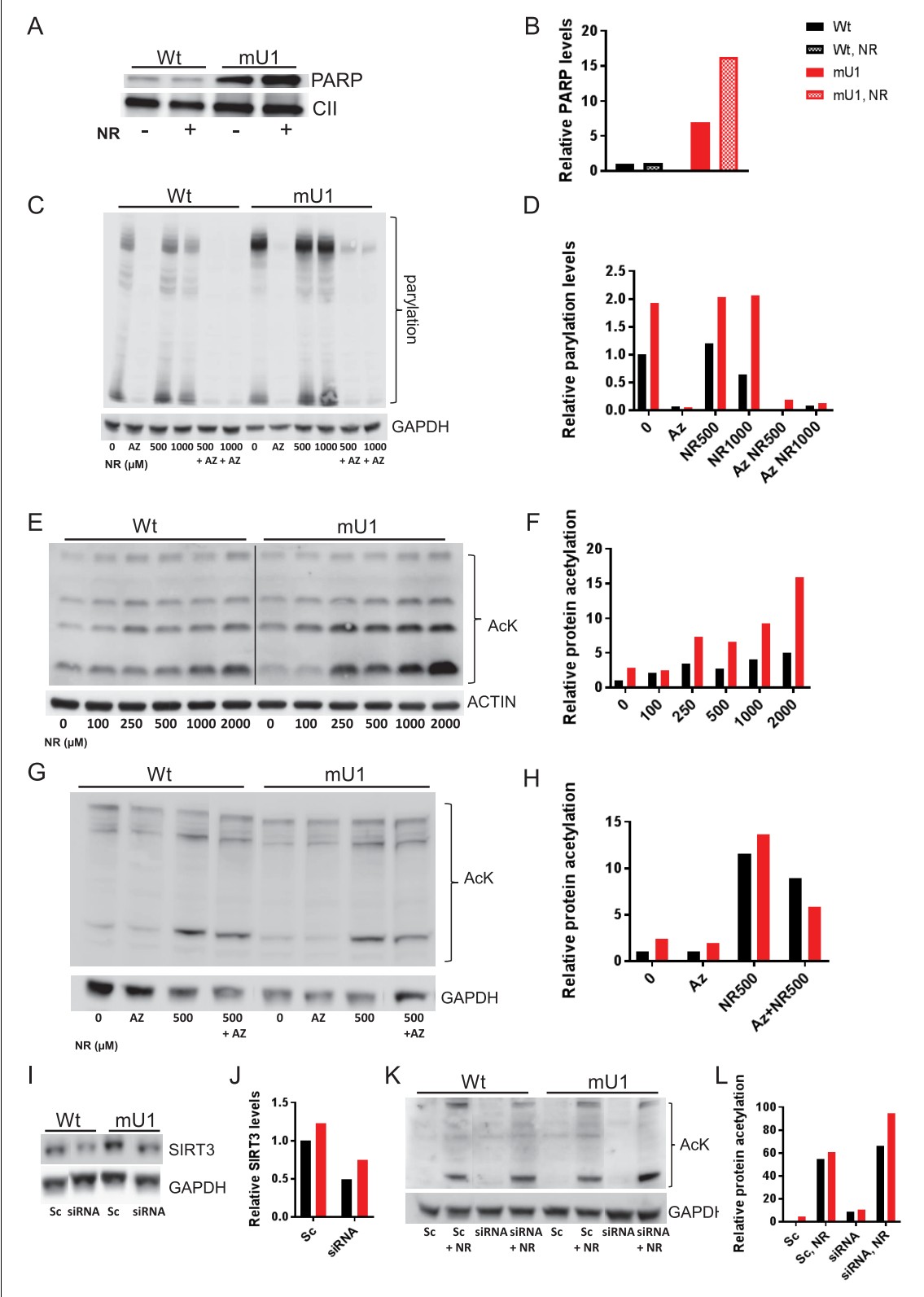

**Figure 6.** Increasing doses of nicotinamde riboside (NR) progressively inhibit deacetylation and poly(ADP-ribose) polymerase (PARP) cleavage. (A) Western blot of PARP levels in extract from cells with (mU1) and without (wt) expression of mutUNG1 grown in vitro with or without NR. (B) Quantification of PARP levels in western blot. (C) Western blot of poly:mono-ADP ribose/PARylation levels in cells with (mU1) and without (wt) expression of mutUNG1, treated with PARP inhibitor and/or NR. (D) Quantification of PARylation in western blot. (E) Western blot of protein acetylation

*Figure 6 continued on next page*

*Figure 6 continued*

levels in total extract in cells with (mU1) and without (wt) expression of mutUNG1, and increasing concentrations of NR. (F) Quantification of protein acetylation levels in western blot. (G) Western blot of acetylation levels in cells with (mU1) and without (wt) expression of mutUNG1, treated with PARP inhibitor and/or NR. (H) Quantification of acetylation in western blot. (I) Western blot of SIRT3 levels in cells with (mU1) and without (wt) expression of mutUNG1, transfected with SIRT3 silencing RNA (siRNA) and scrambled control. (J) Quantification of SIRT3 levels in western blot. (K) Western blot of acetylation levels in cells with (mU1) and without (wt) expression of mutUNG1, transfected with SIRT3 siRNA and scrambled control and NR treatment. (L) Quantification of acetylation in Western blot. Abbreviations: Az=AZD2461/Olaparib, Sc = Scrambled. Raw data are presented in *Source data 2*.

*Wellman et al., 2019*). In addition, *SIRT3* transcription is stimulated by ROS, and the increased levels of SIRT3 might be a reflection of elevated ROS production in mutUNG1-expressing mice (*Lauritzen et al., 2015*). Since SIRT3 regulates the activity of a number of molecular processes, including the antioxidant function of SOD2, a loss of SOD2 activation, as shown in mitochondrial extract from mutUNG1-expressing cardiac tissue, would lead to a further increase in mtDNA damage and ROS production, causing a negative spiral that would be detrimental to the cell (*Chen et al., 2011*). Additionally, SIRT3 controls OPA1 activity through deacetylation, and is thereby an important part in the regulation of mitochondrial structure and function (*Samant et al., 2014*). The lack of activity and control of OPA1 could cause dysfunction of overall mitochondrial homeostasis in cardiac tissue, and illustrates the importance of functional SIRT3 levels in cardiomyocytes. Interestingly, it was recently shown that adjacent cardiomyocyte mitochondria exhibit sites of membrane contact and inter-organelle alignment of cristae structures, suggesting a means of intracellular communication (*Picard et al., 2015*). Owing to the central role of OPA1 in maintaining inner mitochondrial structure, reduced OPA1 activity due to loss of acetylation could therefore cause severe cardiac impairment by inducing mitochondrial dysfunction, which is exemplified in the mutUNG1-expressing mice. In the present study, we confirm our previous data on a disorganized mitochondrial structure in the mutUNG1 myocardium, and we now show that this disorganization most likely involves (de) acetylated OPA1. Our findings underscore the role of intact mitochondrial organization for mitochondrial function and point to a possible mechanism for the harmful effect of cardiac mitochondria in mutUNG1-expressing mice.

NR has recently become a popular tool for boosting NAD$^+$ levels, and has been reported to have anti-aging properties and suggested as a treatment for patients suffering from brain disorders including Alzheimer's disease (*Gong et al., 2013*). However, in our study, NR does not seem to have overall positive effects. On the contrary, NR supplementation induced a failing of cristae structures to align between neighboring mitochondria and the IMJ sites seemed to be lost. The metabolic dynamics are tightly connected to mitochondrial structure (*Picard et al., 2015*), and F1Fi-ATP synthase, the mitochondrial contact site and cristae organizing system (MICOS), and OPA1 are three membrane-shaping components exhibiting crucial roles in cristae biology. NR supplementation induced metabolic changes as shown by altered protein levels for proteins involved in FA (fatty acid) metabolism, TCA cycle, and OXPHOS (oxidative phosphorylation), including components of F1Fi-ATP synthase. Further, Mic60, -19, and -26 of the MICOS complex were all upregulated. Upregulation of these proteins could be a compensatory mechanism initiated by the metabolic stress. Interestingly, a recent published article found increased levels of Mic60 when knocking out OPA1 (*Stephan et al., 2020*). Intriguingly, these mitochondria lacking OPA1 had a similar appearance to NR-treated Wt mitochondria, with a circular shape and less ordered cristae structure. Additionally, also in line with fragmented mitochondrial structure, proteomic analysis showed that both mutUNG1-expressing and Wt mice treated with NR had increased levels of mitochondrial fission one protein (FIS1), an important regulator of mitochondrial fission (*Samant et al., 2014*).

The consequence of this might be more severe in cardiac than in neuronal tissue, since mitochondria behave and communicate in a completely different manner in the two tissues. Whereas in neurons the mitochondria move quite dynamically in the cell (*Tang et al., 2019*), cardiac mitochondria are much more fixed in the cell within the myofibril fiber lattice (*Picard et al., 2015*). This means that the contact points between the mitochondria, where inter-organelle communication takes place, are particularly important in cardiac tissue. If NR thereby causes the cardiac mitochondria to change shape in such way that they lose this inter-organelle communication, this could prove harmful for the heart. Based on our data, one should therefore use NR with care clinically, and it is important to obtain more information about the correct dose that should be used. More importantly, if the

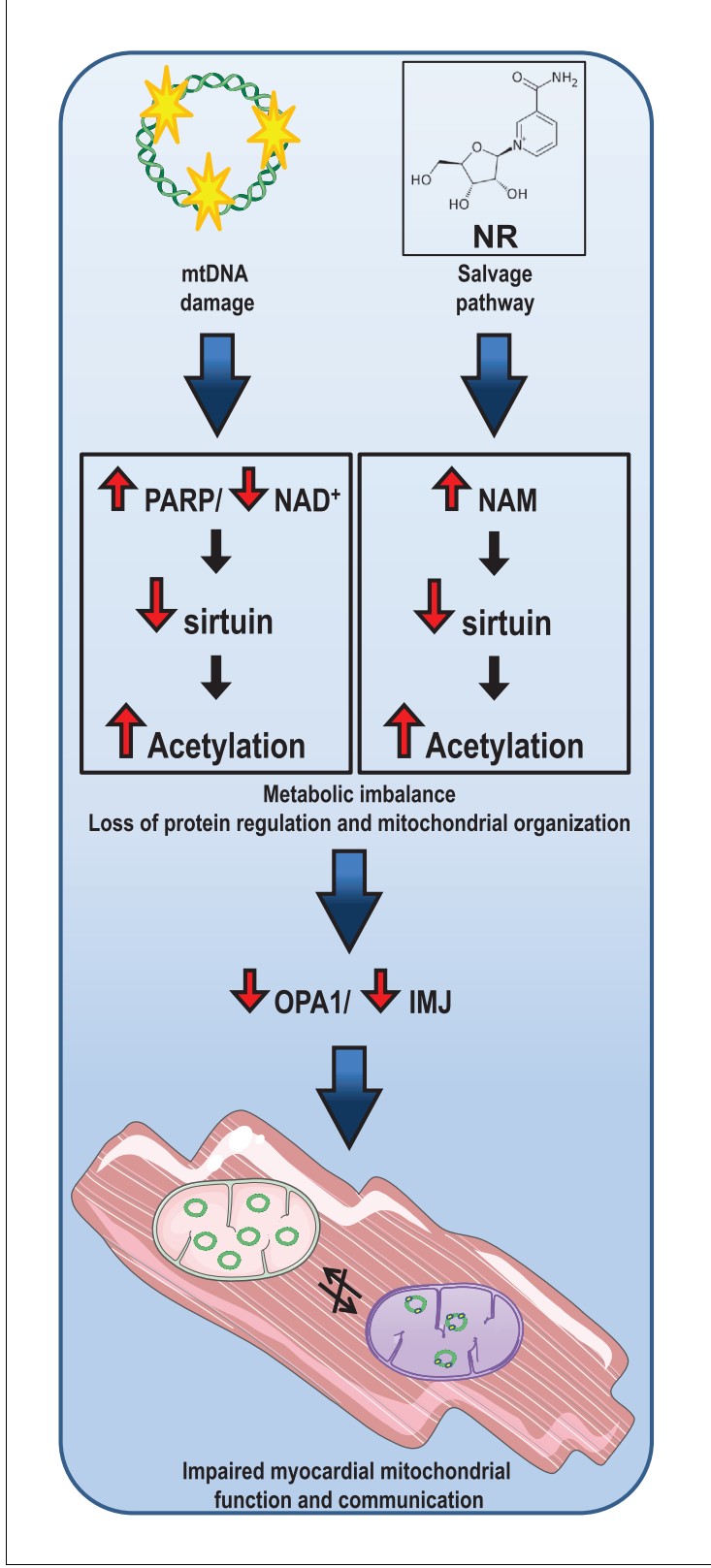

**Figure 7.** Illustration of the concept behind effect of loss of nicotinamide adenine dinucleotide (NAD$^+$) due to overactive (mt). DNA damage repair and consequences of nicotinamide (NAM) accumulation in cardiac tissue.

patient is believed to have cardiomyopathies caused by mitochondrial dysregulation and mtDNA irregularities, for example, by using anthracycline regiments (*Lebrecht and Walker, 2007*), extra care of NR supplementation should be considered.

The present study has some limitations. Not all findings were statistically significant, and these data must be interpreted with caution. Association does not necessarily mean any causal relationship and additional experimental studies are needed to fully support our conclusion. Moreover, the relevance of these studies to human pathology is still uncertain.

In conclusion, high levels of mtDNA damage can cause a drop in $NAD^+$ levels through increased PARP activity, which ultimately causes a breakdown of function and communication in cardiac mitochondria. Treatment with $NAD^+$ precursors such as NR might not get the desired effect of re-establishing the activity of $NAD^+$-dependent proteins due to irregularities in the $NAD^+$ metabolism, which can cause negative feedback mechanisms, including high levels of the sirtuin inhibitor NAM. This may lead to disturbances in mitochondrial and cellular pathways including a disorganization of mitochondrial structure. Our data suggest that the use of NR in cardiac disorders should be examined more closely, in particular at higher dosages.

# Materials and methods

## Key resources table

| Reagent type (species) or resource | Designation | Source or reference | Identifiers | Additional information |
|---|---|---|---|---|
| Genetic reagent (*Mus musculus*) | mutUNG1 | PMID:26055793 | | Dr Knut H Lauritzen (Oslo University Hospital) |
| Cell line (*Homo sapiens*) | HeLA/ mutUNG1 | PMID:20065039 | Cat:631183 RRID:CVCL_V353 | Stable transfected; HeLa Tet-On 3 G Cell Line |
| Chemical compound, drug | Nicotinamide Riboside Chloride (Nigagen) | ChromaDex | Cat:ASB-00014315–15 | |
| Chemical compound, drug | AZD2461/ Olaparib | Merck | Cat:SML1858 | 25 µM |
| Transfected construct (human) | siRNA to SIRT3 (ID:S23766) | Thermo Fisher Scientific | Cat:4392420 | Transfected construct (human) |
| Transfected construct (human) | siRNA, Negative control | Thermo Fisher Scientific | Cat:AM4635 | |
| Antibody | PARP (Rabbit polyclonal) | Cell Signaling | Cat:9542 RRID:AB_2160739 | WB (1:1000) |
| Antibody | SIRT3 (Rabbit polyclonal) | Cell Signaling | Cat:5490 RRID:AB_10828246 | WB (1:1000) |
| Antibody | SDHA (CII) (Mouse monoclonal) | Abcam | Cat:ab14715 RRID:AB_301433 | WB (1:5000) |
| Antibody | Acetylated-Lysine (Rabbit polyclonal) | Cell Signaling | Cat:9441 RRID:AB_331805 | WB (1:1000) |
| Antibody | SOD2 (acetyl K68) (Rabbit monoclonal) | Abcam | Cat:ab137037 RRID:AB_2784527 | WB (1:1000) |
| Antibody | SOD2 (Rabbit polyclonal) | Cell Signaling | Cat:13194 RRID:AB_2750869 | WB (1:1000) |
| Antibody | Poly/Mono-ADP Ribose (E6F6A) (Rabbit polyclonal) | Cell Signaling | Cat:83732 RRID:AB_2749858 | WB (1:1000) |
| Antibody | PHD (Rabbit polyclonal) | Cell Signaling | Cat:2784 RRID:AB_2162928 | WB (1:1000) |

*Continued on next page*

*Continued*

| Reagent type (species) or resource | Designation | Source or reference | Identifiers | Additional information |
|---|---|---|---|---|
| Antibody | OPA1 (Rabbit polyclonal) | Cell Signaling | Cat:80471 RRID:AB_2734117 | WB (1:1000) |
| Antibody | beta-Actin (Mouse monoclonal) | Sigma-Aldrich | Cat:A5441 RRID:AB_476744 | WB (1:1000) |
| Antibody | MFN2 (Mouse monoclonal) | Abcam | Cat:ab56889 RRID:AB_2142629 | WB (1:1000) |
| Antibody | DRP1 (Rabbit polyclonal) | Cell Signaling | Cat:8570 RRID:AB_10950498 | WB (1:1000) |
| Antibody | PGC-1a (Rabbit polyclonal) | Abcam | Cat:ab54481 RRID:AB_881987 | WB (1:1000) |
| Antibody | Anti-Rabbit IgG | Cell Signaling | Cat: 7074 RRID:AB_2099233 | WB: (1:20000) |
| Antibody | Ant-Mouse IgG | Cell Signaling | Cat:7076 RRID:AB_330924 | WB: (1:20000) |
| Sequence-based reagent | NAMPT_f | Merck | PCR primers | ATCCAGGAGG CCAAAGAAG |
| Sequence-based reagent | NAMPT_r | Merck | PCR primers | ATCGGGAGATG ACCATCGTA |
| Sequence-based reagent | NMNT1_f | Merck | PCR primers | TGCATGCTACA GGAAAATAC |
| Sequence-based reagent | NMNT1_r | Merck | PCR primers | AAGTTCTGCC ATGATGATTC |
| Sequence-based reagent | NMNT2_f | Merck | PCR primers | GGCAGATATGG AAGTGATTG |
| Sequence-based reagent | NMNT2_r | Merck | PCR primers | GGAGTATGGAG GAGTGATTC |
| Sequence-based reagent | NMNT3_f | Merck | PCR primers | CAGCATGAAG AACCGAATC |
| Sequence-based reagent | NMNT3_r | Merck | PCR primers | TGGTACCTTCC TGTTTGG |
| Sequence-based reagent | 18 s_f | Merck | PCR primers | CGCGGTTCTAT TTTGTTGGT |
| Sequence-based reagent | 18 s_r | Merck | PCR primers | AGTCGGCA TCGTTTATGGTC |
| Sequence-based reagent | mtDNA_f | Merck | PCR primers | CCCAGCTACTAC CATCATTCAAGT |
| Sequence-based reagent | mtDNA_r | Merck | PCR primers | GATGGTTTGGGAG ATTGGTTGATGT |
| Sequence-based reagent | OGG1_f | Merck | PCR primers | ATGAGGACCAA GCTAGGTGAC |
| Sequence-based reagent | OGG1_r | Merck | PCR primers | GCCTCACAATC AACTTATCCC |
| Commercial assay or kit | Pierce BCA Protein Assay Kit | Thermo Fisher Scientific | Cat:23228 | |
| Commercial assay or kit | RNeasy Mini Kit | Qiagen | Cat:74106 | |
| Commercial assay or kit | Acetyl-Coenzyme A Assay kit | Merck | Cat:MAK039 | |
| Other | RM1+ 6000 ppm | Special Diets Services | | Custom Dox diet |
| Other | TRI Reagent | Merck | Cat:9424 | |
| Other | qScript cDNA Supermix | Quantbio | Cat:95048 | |

*Continued on next page*

*Continued*

| Reagent type (species) or resource | Designation | Source or reference | Identifiers | Additional information |
|---|---|---|---|---|
| Other | PerfeCTa SYBR Green Supermix | Quantbio | Cat: 95054 | |
| Other | Sodium succinate dibasic hexahydrate | Merck | Cat:S2378 | 10 mM |
| Other | Cytochrome c | Merck | Cat:C7752 | 10 µM |
| Other | Halt Protease and Phosphatase Inhibitor | Thermo Fisher Scientific | Cat:1861284 | |
| Other | M-PER Mammalian protein extraction reagent | Thermo Fisher Scientific | Cat:78505 | |
| Other | Dynabeads Protein G | Thermo Fisher Scientific | Cat:10003D | |
| Other | Oligofectamine Reagent | Thermo Fisher Scientific | Cat:12252–011 | |

## Experimental models and treatment

The mutUNG1 mouse model (mU1) was designed and characterized as previously described. Here, a transgene consisting of a human version of Uracil-DNA glycosylase 1 (Ung1) (HGNC ID:12572) with a substitution mutation (Tyr147Ala) has been introduced (*Lauritzen et al., 2015*). Expression of mutUNG1 was induced by addition of doxycycline to the diet corresponding to 6 mg doxycycline/g chow (manufactured by Special Diets Services) when the mice were 8 weeks of age. The mice were sacrificed after 8 weeks of mutUNG1 induction. NR (Niagen from Chromadex) was added to the diet, either as 400 mg/kg chow (medium dose) or 1000 mg/kg chow (high dose), after 6 weeks of mutUNG1 induction (with a total of 2 weeks of NR treatment). These numbers were chosen based on the parameters that a mouse in average weighs 30 g, and eat 3 g chow per day, and published experiments performed by other groups (*Gariani et al., 2016*; *Zhang et al., 2016*). All experiments had three to five animals per group unless stated otherwise, with several biological repeats. All experimental procedures were approved by the Section for Comparative Medicine at the University Hospital of Oslo and by the Norwegian Animal Research Authority, FOTS: 8594 and conducted according to the laws and regulation on animal welfare in Norway and in the European Union.

The cell model was designed and characterized as previously described. Briefly, HeLa Tet-On 3 G Cell Line (Clontech/Takara Bio) was stably transfected with the same transgenic element described for the mice model above. The presence of the transgene was confirmed by sequencing. The cells were incubated in a sterile environment and routinely checked to ensure negative mycoplasma contamination (*Lauritzen et al., 2010*). For PARP inhibition, typically 25 µM Olaparib/AZD2461 (Merck) was used. For SIRT3 knockdown, SIRT3 siRNA (ID: s23766) and negative/scramble control (AM4635) (Thermo Fisher Scientific) was transfected into a six-well plate with the cells, using Oligofectamine Reagent (12252–011, Thermo Fisher Scientific) according to the manufacturer's recommendations for 48 hr before harvest.

## Protein extract preparation and western blotting

Mitochondria were isolated from cardiac tissue by the following protocol: Left ventricular cardiac tissue was briefly washed in ice-cold PBS, transferred to 2 mL buffer A1 (250 mM sucrose, 0.5 mM $Na_2EDTA$, 10 mM Tris, pH 7.4) + 200 µL trypsin (Sigma #T4049), homogenized using a tissue homogenizer, and incubated on ice for 20 min; 2 mL buffer B1 (buffer A1 + 0.1% bovine serum albumin) was added, and the sample was further homogenized by two strokes with a Douncer B and centrifuged at $600 \times g$ at 4°C for 10 min. The supernatant (containing mitochondria) was transferred to a new tube and centrifuged at $9800 \times g$ at 4°C for 10 min. The supernatant was discarded and the pellet was resuspended in 500 µL buffer B1 and centrifuged at $9600 \times g$ at 4°C for 10 min. The supernatant was discarded and the pellet was resuspended in 500 µL buffer B1 and centrifuged at $9200 \times g$ at 4°C for 10 min. The supernatant was discarded and mitochondrial protein was extracted with M-PER Mammalian Protein Extraction Reagent (Thermo Fisher Scientific) containing Halt

Protease and Phosphatase Inhibitor. The purity of the mitochondrial extracts was validated using proteomic analysis and bioinformatics evaluation.

Total protein was extracted by M-PER Mammalian Protein Extraction Reagent or T-PER Tissue Protein Extraction Reagent (Thermo Fisher Scientific) containing Halt Protease and Phosphatase Inhibitor.

Protein concentration was determined by Pierce BCA Protein Assay Kit (Thermo Fisher Scientific). Immunoprecipitation was performed using Dynabeads Protein G (Thermo Fisher Scientific) according to the manufacturer's recommendations.

The protein was separated by SDS-PAGE and transferred to PVDF membrane.

The membranes were developed with Radiance Plus Substrate (Azure Biosystems). The images were captured by LAS-4000 (Fujifilm) and quantified by Image Studio Lite (version 5.2, Li-Cor, Lincoln, NE). All protein quantification blots are normalized against the loading control (CII, PHD, GAPDH or ACTIN).

## HPLC sample preparation and measurement

$NAD^+$ and NAM extraction and subsequent quantitative analysis were performed as previously described (*Yoshino and Imai, 2013*). Briefly, tissue sample (typically 50–70 mg) was homogenized in 500 μL $HClO_4$ (0.4 M), incubated on ice for 5 min and precipitated by adding 80 μL KOH (0.2 M) with shaking for 2 min. The samples were then centrifuged at $3000 \times g$, 10 min, 4°C. Supernatant was filtered using Spin-X centrifuge tube (Costar, filter size 0.22 μM) and the samples were stored in −80°C until HPLC measurements. The samples from tissues were subjected to HPLC using a 20 mm × 3.9 mm Sentry Guard column (Nova-Pak C18 bonded silica) connected to a 150 mm × 4.6 mm Atlantis T3 silica-based, reversed-phase C18 columns (Waters Corporation). $NAD^+$ and NAM were detected by UV detector and UV absorbance was monitored at 261 nm. Elution of $NAD^+$ and NAM from samples was verified and quantified by co-elution with known amounts of $NAD^+$ and NAM standards (Sigma-Aldrich).

## Echocardiography

Examination was performed with the VEVO 2100 system (VisualSonics, Toronto, Canada). Mice were lightly anesthetized with a mixture of 98.25% $O_2$ and 1.75% isoflurane maintained by mask ventilation and were placed on a heated examination table to maintain body temperature. Standard echocardiography examination, including long and short axis images of the left ventricular (LV) and atrium, was performed (*Finsen et al., 2005*). Recorded data were analyzed offline using the Vevo LAB 3.2 software (VisualSonics). Data were assembled from 8 to 16 animals per group, pooled results of several biological repeats. Relative LV wall thickness was calculated with the formula: (2*LVPW:d)/(LVID:d).

## Proteomic analysis

Cardiac mitochondria were isolated from 15 tissue samples as explained above, and lysed in M-PER buffer (Thermo Fisher Scientific) with 0.5% Triton X-100, and aliquoted to 100 μg protein. The proteins were precipitated with TCA/acetone in −20°C overnight. The precipitated proteins were dissolved with 6 M urea in 50 mM ammonium bicarbonate, reduced with DTT, and alkylated with iodoacetamide. Then, the proteins were in-solution digested by diluting the urea concentration to 1 M followed by digestion with trypsin overnight in 37°C. The resulting peptides were desalted and concentrated before mass spectrometry by the STAGE-TIP method using a C18 resin disk (3M Empore). Each peptide mixture was analyzed by an nEASY-LC coupled to QExactive Plus (Thermo-Electron, Bremen, Germany) with EASY Spray PepMapRSLC column (C18, 2 μL, 100 Å, 75 μM × 50 cm) using a 60 min LC separation gradient. The resulting MS raw files were submitted to the MaxQuant software version 1.6.1.0 for protein identification and label-free quantification. Carbamidomethyl (C) was set as a fixed modification and acetyl (K), carbamyl (N-term), and oxidation (M) were set as variable modifications. First search peptide tolerance of 20 ppm and main search error of 4.5 ppm were used. Trypsin without proline restriction enzyme option was used, with two allowed miscleavages. The minimal unique+razor peptides number was set to 1, and the allowed FDR was 0.01 (1%) for peptide and protein identification. Label-free quantitation was employed with default settings. The Uniprot database with 'mouse' entries (January 2019) was used for the database searches.

Perseus software version 1.6.1.3 was used for the statistical analysis of the results. Known contaminants as provided by MaxQuant and identified in the samples were excluded from further analysis. Metascape and Pheatmap R package 1.0.12 was used for further data analysis (R: A language and environment for statistical computing. R Foundation for Statistical Computing, Vienna, Austria. URL https://www.R-project.org/.pheatmap. Pheatmap: Pretty Heatmaps. R package version 1.0.12. https://CRAN.R-project.org/package=pheatmap).

## Electron microscopy

Pieces of heart tissue were immersion fixed in a phosphate buffered solution containing 1% paraformaldehyde and 2.5% glutaraldehyde. The samples were then dissected into small rectangular pieces (~0.5 mm × 0.5 mm × 1 mm) and cryoprotected by immersion in graded concentrations of glycerol (10%, 20%, and 30%) in 0.1 M phosphate buffer (75 mM $Na_2HPO_4$ and 25 mM $NaH_2PO_4$, pH 7.4). The samples were plunged into liquid propane cooled to −170°C by liquid nitrogen in a Universal Cryofixation System KF80 (Reichert-Jung). For freeze substitution (Bergersen et al., 2008), the samples were immersed in a solution of anhydrous methanol and 0.5% uranyl acetate overnight at −90°C. The temperature was then raised stepwise in 4°C increments per hour from −90°C to −45°C, where it was kept for the subsequent steps. The tissue samples were washed several times with anhydrous methanol to remove residual water and uranyl acetate, and then infiltrated in the embedding resin Lowicryl HM20 stepwise from Lowicryl/methanol 1:2, 1:1, and 2:1 (1 hr each) to pure Lowicryl (overnight). Polymerization was catalyzed by 360 nm ultraviolet light for 2 days at −45°C followed by 1 day at room temperature. Ultrathin sections (70 nm) were cut by a diamond knife on a Reichert-Jung ultramicrotome and mounted on nickel grids with an adhesive pen (David Sangyo). The ultrathin sections were contrasted in uranyl acetate (5%) and lead citrate (30%), before they were observed in a Philips CM100 EM.

## Analysis and quantification of EM micrographs

Cristae organization and cristae inter-organelle organization was evaluated based on a scoring system where 5 was the highest level of organization and 1 the lowest. For mitochondrial shape/circularity, scoring was also applied but 5 was the highest degree of circularity and 1 the lowest. One to five images were evaluated for each animal, N=5 for each group except Wt where N=4.

IMJ length was quantified using ImageJ, where total measured IMJ length was normalized against total mitochondrial area. N=5.

All analyzed samples were from area LV tissue with myocardium cut in longitudinal direction. All samples were analyzed, randomized, and blinded.

## qPCR

Total RNA was isolated from mouse hearts, using RNeasy Tissue Mini Kit (Qiagen) in combination with Trizol Reagent (Sigma-Aldrich, Merck, Darmstadt, Germany) in accordance with the manufacturer's recommendations. cDNA was produced from the isolated RNA using qScript cDNASupermix (Quantabio). For rt-PCR measurement of mtDNA copy number, total DNA was isolated using Dneasy Blood and Tissue Kit (Qiagen). rt-PCR reactions were carried out in a 20 μL mixture containing PerfeCTa SYBR Green Supermix (Quantabio), 100 nM of each primer, and 10 ng cDNA or 5 ng total DNA. All reactions were done in triplicates. Negative controls with water were performed for each target. Standard curves with a 5-point 1:10 dilution series, starting with 100 ng, were made for each target. Default PCR program settings were used. All reactions were run on a Stratagene Mx3005P (Agilent Technologies) using the default settings recommended by the manufacturer and analyzed using MxPro software. Data were calculated based on the standard curves (standard curve method), and target of interest was normalized against the control target gene(S) (18 s for cDNA and OGG1 for mtDNA). Standard curves with $R^2$ values of <0.99 were rejected.

## Oxygraph

Mitochondrial complex II-driven respiratory capacity measured in heart homogenates were analyzed by high-resolution respirometry (Oxygraph-2K; Oroboros). Frozen heart tissue (left ventricle) was homogenized in 1 mL of MSHE buffer, pelleted at 10,000 × g, washed once, and resuspended in 250 μL of MSHE buffer. Mir05 buffer was supplied with succinate (10 mM). Cytochrome C (10 μM)

was added when respiration was stabilized. The succinate-/cytochrome C-based respiration was then set as respiration capacity.

## Acetyl CoA levels
Acetyl CoA levels were measured using Acetyl-Coenzyme A Assay kit (Sigma-Aldrich, Merck, Darmstadt, Germany) according to the manufacturer's recommendations.

## Statistics
All experiments using animals were n=3–5 per group unless stated otherwise, and several biological repeats were conducted for all experiments. All quantitative data are represented as means ± the standard error of the mean. Unpaired, two-tailed t-tests were performed unless otherwise stated. The null hypothesis was rejected at the 0.05 level.

## Acknowledgements
We are grateful to Chromadex Inc for generously providing NR to this study. We are also grateful to the Norwegian Research Council, the Helse Sør-Øst Regional Health Authority of Norway, and the National Association for Public Health ('Nasjonalforeningen for folkehelsen') for financial support of this work. We would also like to thank Hilde Loge Nilsen and Jon Storm-Mathisen for their help with this manuscript. We used images from https://smart.servier.com/ to create the graphical abstract.

## Additional information

### Funding

| Funder | Author |
| --- | --- |
| Nasjonalforeningen for Folke-helsen | Knut H Lauritzen |
| Helse Sør-Øst RHF | Knut H Lauritzen<br>Pål Aukrust<br>Arne Yndestad |
| Norges Forskningsråd | Bente Halvorsen |

The funders had no role in study design, data collection and interpretation, or the decision to submit the work for publication.

### Author contributions
Knut H Lauritzen, Conceptualization, Data curation, Software, Formal analysis, Validation, Investigation, Visualization, Methodology, Writing - original draft, Project administration, Writing - review and editing; Maria Belland Olsen, Data curation, Formal analysis, Validation, Visualization, Writing - review and editing; Mohammed Shakil Ahmed, Data curation, Investigation, Methodology, Writing - review and editing; Kuan Yang, Data curation, Software, Formal analysis, Validation, Investigation, Visualization, Writing - review and editing; Johanne Egge Rinholm, Investigation, Writing - review and editing; Linda H Bergersen, Håvard Attramadal, Resources, Methodology, Writing - review and editing; Qin Ying Esbensen, Lars Jansen Sverkeli, Mathias Ziegler, Methodology, Writing - review and editing; Bente Halvorsen, Resources, Funding acquisition, Writing - review and editing; Pål Aukrust, Conceptualization, Resources, Supervision, Funding acquisition, Validation, Writing - original draft, Project administration, Writing - review and editing; Arne Yndestad, Conceptualization, Resources, Supervision, Funding acquisition, Validation, Project administration, Writing - review and editing

### Author ORCIDs
Knut H Lauritzen (iD) https://orcid.org/0000-0002-6003-6027
Kuan Yang (iD) https://orcid.org/0000-0002-2246-4864
Johanne Egge Rinholm (iD) https://orcid.org/0000-0003-3741-850X

Mathias Ziegler  https://orcid.org/0000-0001-6961-2396
Bente Halvorsen  https://orcid.org/0000-0002-6529-6485

### Ethics

Animal experimentation: All experimental procedures involving animals were approved by the Section for Comparative Medicine at the University Hospital of Oslo and by the (Norwegian Animal Research Authority, FOTS: 8594) and conducted according to the laws and regulation on animal welfare in Norway and in the European Union.

### Decision letter and Author response

Decision letter https://doi.org/10.7554/eLife.59828.sa1
Author response https://doi.org/10.7554/eLife.59828.sa2

## Additional files

### Supplementary files

- Source data 1. Rawdata; protemics.
- Source data 2. Rawdata; all quantifications.
- Source data 3. Rawdata; all supplementary quantifications.
- Source data 4. Raw data; original western blots.
- Supplementary file 1. Table of gene ontology (GO) terms from proteomic analysis.
- Transparent reporting form

### Data availability

Datasets associated with this article are provided as source data.

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
