## [Decision Letter]

**Acceptance summary:**

Overall, this is an interesting report with the potential to shed light on the failure of NAD+ supplementation treatments in the context of diseases that result in bioenergetic defects. Surprisingly, and in contrast to results from other groups using other models of heart failure, the study shows that supplementing NAD with nicotinamide riboside does not improve cardiac function or NAD content in their transgenic mouse model expressing a mutation in a mitochondrial DNA repair enzyme in heart.

**Decision letter after peer review:**

Thank you for submitting your article "Instability in NAD + metabolism leads to impaired cardiac mitochondrial function and communication" for consideration by *eLife*. Your article has been reviewed by 3 peer reviewers, including Tinna V Stevnsner as the Reviewing Editor and Reviewer #1, and evaluation has been overseen by Didier Stainier as the Senior Editor. The following individual involved in review of your submission has agreed to reveal their identity: Joseph A Baur (Reviewer #3).

The reviewers have discussed the reviews with one another and the Reviewing Editor has drafted this decision to help you prepare a revised submission.

Summary:

In this manuscript, Lauritzen et al., report that, in a transgenic mouse model expressing a mutation in a mitochondrial DNA repair enzyme in heart, prolonged PARP activation during mtDNA repair is associated with reductions in NAD+, a cofactor necessary for PARP activity.

The authors also show that, in this context, sirtuins activity is reduced as a potential consequence of PARP-induced NAD+ reductions, resulting in subsequent elevations in protein acetylation. Given the role of acetylation in the regulation of key mitochondrial proteins, such as OPA1, the authors argue that these alterations in acetylation could be behind some of the mitochondrial structural defects shown in their mtUNG1 mouse model. Surprisingly, and in contrast to results from other groups using other models of heart failure, supplementing NAD with NR does not improve cardiac function or NAD content. Moreover, it alters mitochondrial morphology in wt hearts and at a high dose, increases acetylation of mitochondrial proteins in mutant hearts, and can induce mitochondrial responses in liver tissues, and further sirtuin inhibition by the accumulation of NAM.

Overall, this is an interesting report with the potential to shed light into the failure of NAD+ supplementation treatments in the context of diseases that result in bioenergetic defects. The manuscript is in general well written. However, before the manuscript is acceptable for publication, a major revision is required were several points need to be addressed and substantial new data is required.

– The author's conclusions are not fully supported by the data. Most of the results presented are essentially association of phenotypes. Thus, rescue experiments are necessary to further support the authors' statements.

– It is well known that the metabolism of NAD+ is extremely complex and linked to major energetic pathways, including glycolysis, TCA, FA oxidation, ketone body metabolism, and mitochondrial oxidative phosphorylation. Therefore, reductions in NAD+ in the context of mutUNG1 can be due to mitochondrial respiratory defects, and/or alterations in glycolysis, TCA cycle, or even in the function of NADH/NAD+ transporters. Therefore, the authors need to test for the contribution of these pathways in their models. The authors should also contemplate the fact that redox reactions are readily reversible, and they do not contribute directly to changes in the total NAD pool in a specific subcellular compartment. NAD+ and NADH interconvert but are not irreversibly consumed.

Essential revisions:

1) There is some question of the dosing used as the authors state "400 mg/kg food/day" but cite papers that use 400 mg/kg of mouse/day (i.e., ~7 fold higher dosing). It is presumed that the dose was actually based on body weight after reading the wording in the methods, but if this is not the case, it needs to be clarified and a higher dose should be examined.

2) In Figure 1 the authors show the association between increases in PARP, reductions in NAD+ and alterations in sirtuin activity. This association does not prove causation. A series of rescue experiments are necessary to support a direct causal link between these molecular events. (e.g. silencing or inhibition of PARP and rescue acetylation and/or mitochondrial morphology).

Likewise, alterations in the acetylation of OPA1 do not suffice to link this defect to the alterations in mitochondrial morphology observed in mutUNG1 tissues. Please check for MFN2 levels or DRP1 localization, expression of PGC1- α, and/or ratio of mtDNA/ nDNA in the conditions and tissues under study.

Could other organs compensate for the loss of cardiac NAD+?

3) The model proposed is that PARP1 localized to mitochondria senses mtDNA damage and depletes mitochondrial NAD. Yang et al. (Cell 103: 1095, 2007) suggests that PARP1 activation spares mitochondrial NAD. In addition, numerous studies have demonstrated that damaged mtDNA can be released into the cytosol and circulation – how is the alternative hypothesis excluded that damaged mtDNA is released and activates nuclear PARP1? Is PARylation increased in nuclear extracts? Is acetylation? The evidence that the entire system is taking place inside the mitochondria is very sparse even though this conclusion is heavily implied. The authors at least need to demonstrate that their mitochondrial extracts are free of nuclear/cytosolic proteins.

4) Please, check acetylation levels in WT liver after high dose of NR supplementation to understand the association between NAM levels and sirtuin inhibition. All acetylation changes are attributed to SIRT3, but this is only one part of the equation – what about the "on" rate for acetylation? Prior studies have identified SIRT3 depended and independent sites. Are the SIRT3-independent sites not affected?

While it is beyond the scope of the present study to cross to SIRT3 deficient mice, the cell culture system provides an opportunity to test SIRT3-dependence. SIRT3 should be knocked down or deleted with CRISPR in this system and it should be tested whether NR still influences acetylation. What happens to acetyl-CoA levels?

5) The authors propose that reductions in NAD+ can result into mitochondria defects via sirtuin inactivation. However, NAD supplementation does not rescue mitochondrial morphological defects in mutant tissues, nor it rescues sirtuin activity. To clarify this point, the authors should measure mitochondrial respiration after NAD supplementation.

6) The implication that the NR effects on mitochondria in wt hearts are detrimental is based mainly on morphology. Is there evidence for a defect in respiration or bioenergetics? Similarly, acetylation is implied to be bad, but several recent studies have questioned the assertion that acetylation impairs mitochondria (e.g., Fisher-Wellman et al., Cell Reports, 26:1557, 2019).

7) The manuscript presents proteomics data that are not discussed. These data shows that under NAD+ supplementation, the expression of specific metabolic enzymes such as carnitine transporters is significantly increased. This points towards an overall metabolic shift after this NR treatment. The authors need to clarify and discuss these data. Moreover, NR supplementation results in the increase of other proteins such as SLC25A46 which has been shown to promote mitochondrial fission. This suggest that other factors are in play that should be addressed.

8) Increases in NAM in heart after high dose of NR supplementation suggest that NAD+ is being metabolized at heart and converted into NAM. However, this point needs to be clarified. Please, report NAD+ levels in heart after high dose of NR supplementation.

To demonstrate causality, inhibition of PARP should revert NAM accumulation.

Accumulation of NAM could be caused by reductions in the salvage pathway or the conversion of NAM into NAD+. Please check NAMPT and NMNAT1-3.

---

## [Author Response]

Essential revisions:1) There is some question of the dosing used as the authors state "400 mg/kg food/day" but cite papers that use 400 mg/kg of mouse/day (i.e., ~7 fold higher dosing). It is presumed that the dose was actually based on body weight after reading the wording in the methods, but if this is not the case, it needs to be clarified and a higher dose should be examined.

In both the cited papers regarding NR doses (Gariani et al., 2016 and Zhang et al., 2016), they write: “NR-treated animals were fed pellets containing vehicle … or NR (400mg/kg/day) for … weeks…”. After carefully reading their manuscript we interpreted this statement as the NR calculations were based on mg/kg food/day (not mg/ kg mouse/day) as we explain in the manuscript. To achieve specific NR doses to the each individual mouse based on weight would acquire other means of NR administration than mixing it in the food, like gavage or injections, which are methods that should be avoided if necessary due to animal welfare and the high level of discomfort. Nonetheless, we have also tested higher doses of NR in the manuscript, especially the experiments regarding stably transfected cells (Figure 6).

Notwithstanding, we apologize for making this part of the manuscript uncertain and this issue has now been more clearly stated in the revised manuscript.

2) In Figure 1 the authors show the association between increases in PARP, reductions in NAD+ and alterations in sirtuin activity. This association does not prove causation. A series of rescue experiments are necessary to support a direct causal link between these molecular events. (e.g. silencing or inhibition of PARP and rescue acetylation and/or mitochondrial morphology).Likewise, alterations in the acetylation of OPA1 do not suffice to link this defect to the alterations in mitochondrial morphology observed in mutUNG1 tissues. Please check for MFN2 levels or DRP1 localization, expression of PGC1- α, and/or ratio of mtDNA/ nDNA in the conditions and tissues under study.Could other organs compensate for the loss of cardiac NAD+?

We appreciate these valuable suggestions regarding increased PARP activity, reduced NAD+ and alterations in sirtuin activity. To accommodate the Reviewer´s important concerns we have now treated the mutUNG1 cell model with the PARP inhibitor olaparib and NR (now Figure 6E-H). Olaparib inhibited parylation/PARP-activity successfully, and the acetylation levels were lower in the groups treated with PARP inhibitor, indicating a link between PARP-activity and deacetylase activity i.e., sirtuin activity.

We have now also checked the protein levels of MFN2, DRP1 and PGC1a (Figure 4 – supplement 1A-C), and we found no differences between mutUNG1 expressing mice and Wt mice, indicating that increased levels of these protein does not play a noticeable part in the mitochondrial morphology observed in mutUNG1 mice. Moreover, there was a decrease in mtDNA copy number in mutUNG1 expressing mice (as seen previously: Lauritzen KH, et al., 2015) and as demonstrated in the new experiment, NR treatment did not increase the mtDNA numbers up to Wt levels (Figure 4 – supplement 2).

All these new experiments, as suggested by the Reviewer, have now been included in the revised manuscript, and we believe that these data more strongly support a link between PARP and Sirtuin activity as an important pathway that are modulated in mutUNG1 mice as well as a role for altered OPA1 acetylation in the altered mitochondrial morphology in these mice.

Finally, we have now also measured protein acetylation levels in the liver with and without NR, and there were no significant differences between the groups suggesting that at least the liver do not compensate for the loss of cardiac NAD+ (Figure 5 – supplement 2). These new data have now been included in the revised manuscript and we have also underscored that we cannot exclude that other tissues could at least partly compensate for the loss of cardiac NAD+.

For clarity, we also chose to remove (originally labeled) Figure 5E-F regarding acetylation of Opa1 in the cell model from the manuscript, since we present no figures/data regarding mitochondrial morphology in this cell line and the cell line was used to substantiate data from the mouse model.

3) The model proposed is that PARP1 localized to mitochondria senses mtDNA damage and depletes mitochondrial NAD. Yang et al. (Cell 103: 1095, 2007) suggests that PARP1 activation spares mitochondrial NAD. In addition, numerous studies have demonstrated that damaged mtDNA can be released into the cytosol and circulation – how is the alternative hypothesis excluded that damaged mtDNA is released and activates nuclear PARP1? Is PARylation increased in nuclear extracts? Is acetylation? The evidence that the entire system is taking place inside the mitochondria is very sparse even though this conclusion is heavily implied. The authors at least need to demonstrate that their mitochondrial extracts are free of nuclear/cytosolic proteins.

To accommodate the Reviewer´s valuable concern, we have performed two additional experiments (Figure 6C-D in the main manuscript), regarding parylation levels.

To evaluate the mitochondrial extracts used in this study we utilized the proteomic analysis and performed a bioinformatic evaluation linking all proteins to cellular compartments by gene ontology (GO) terms. From the 812 proteins, 28 were not linked to a GO term and excluded. From the 784 proteins left, 40 proteins could from their GO terms be connected to nucleus. Among these 40 proteins, 23 proteins did also have a GO-term linking them to mitochondria, leaving 17 proteins connected to nucleus. Most of these could be further linked to the nuclear envelope. As mitochondria interact with other organelles one could expect some contamination of non-mitochondria proteins in these extracts. However, as the level of nuclear proteins were very low (2.2%), and mainly linked to the nuclear membrane, we considered the mitochondrial purity of this extract to be satisfactory. Nuclear proteins detected in this analysis are now listed in Supplementary file 1.

We have in addition included the interesting article that was cited by the Reviewer in the revised reference list, and briefly discussed the import issue that was raised by the Reviewer.

4) Please, check acetylation levels in WT liver after high dose of NR supplementation to understand the association between NAM levels and sirtuin inhibition. All acetylation changes are attributed to SIRT3, but this is only one part of the equation – what about the "on" rate for acetylation? Prior studies have identified SIRT3 depended and independent sites. Are the SIRT3-independent sites not affected?While it is beyond the scope of the present study to cross to SIRT3 deficient mice, the cell culture system provides an opportunity to test SIRT3-dependence. SIRT3 should be knocked down or deleted with CRISPR in this system and it should be tested whether NR still influences acetylation. What happens to acetyl-CoA levels?

Protein acetylation levels were measured in liver with high NR dose (Figure 5 – supplement 2), with similar doses as seen in heart tissue (Figure 5C-D). However, in contrast to the protein acetylation levels in heart tissue, we did not observe significant differences in liver tissue.

Due to technical and experimental issues and current logistical restraints, we could not perform SIRT3 knock down using CRISPR in our lab at this moment. However, we did test SIRT3-dependence by knocking down SIRT3 expression using siRNA technology, which showed more than 50% reduction in SIRT3 protein levels in the cell model (Figure 6I-J). Protein acetylation assays did show a higher degree of protein acetylation in cells treated siRNA for SIRT3 knock down and NR, and this new experiment demonstrates that SIRT3 does indeed influence the acetylation status when high levels of NR are introduced (Figure 6K-L).

Finally, we have now measured acetyl CoA levels in heart tissue from mutUNG1 and Wt mice treated with and without NR, and no effect of NR were found in either of the genotypes (Figure 5 – supplement 4), indicating that acetylation of CoA is not implicated in the differences between the genotypes or in the effect of NR in these mice. These new experiments have now been included in the revised manuscript and briefly commented on.

5) The authors propose that reductions in NAD+ can result into mitochondria defects via sirtuin inactivation. However, NAD supplementation does not rescue mitochondrial morphological defects in mutant tissues, nor it rescues sirtuin activity. To clarify this point, the authors should measure mitochondrial respiration after NAD supplementation.

This is a valid concern. As previously reported by us, there was a decrease in mitochondrial respiration in mutUNG1 expressing mice (as seen previously: Lauritzen KH, et al., 2015), and as shown in the revised manuscript, NR treatment did not increase the respiration up to Wt levels (Figure 4 – supplement 3). Thus, at least NR supplementation, as a tool to increase NAD+ levels, did not rescue the mitochondrial morphology defect or the impaired mitochondrial respiration in mutUNG1 mice. These new data have now been included in the revised manuscript.

6) The implication that the NR effects on mitochondria in wt hearts are detrimental is based mainly on morphology. Is there evidence for a defect in respiration or bioenergetics? Similarly, acetylation is implied to be bad, but several recent studies have questioned the assertion that acetylation impairs mitochondria (e.g., Fisher-Wellman et al., Cell Reports, 26:1557, 2019).

This is an important question and to accommodate the Reviewers concerns we have performed additional experiments. We now show that there is a decrease in mitochondrial respiration in mutUNG1 expressing mice (as seen previously: Lauritzen KH, et al., 2015), and NR treatment did not increase the respiration up to Wt levels (Figure 4 – supplement 3).

7) The manuscript presents proteomics data that are not discussed. These data shows that under NAD+ supplementation, the expression of specific metabolic enzymes such as carnitine transporters is significantly increased. This points towards an overall metabolic shift after this NR treatment. The authors need to clarify and discuss these data. Moreover, NR supplementation results in the increase of other proteins such as SLC25A46 which has been shown to promote mitochondrial fission. This suggest that other factors are in play that should be addressed.

We share this valid concern of the Reviewer, and the proteomic data are now discussed in more detail in the text as a separate section. In addition, this is accompanied with a (now) separate and updated figure (Figure 4). Thus, in the revised manuscript we have included updated versions of the figures interpreting the proteomatic data (Figure 4A), as well as an illustration showing important regulators of the cristae structure, highlighting selected DEPs (differentially expressed proteins) from the dataset (Figure 4B).

8) Increases in NAM in heart after high dose of NR supplementation suggest that NAD+ is being metabolized at heart and converted into NAM. However, this point needs to be clarified. Please, report NAD+ levels in heart after high dose of NR supplementation.To demonstrate causality, inhibition of PARP should revert NAM accumulation.Accumulation of NAM could be caused by reductions in the salvage pathway or the conversion of NAM into NAD+. Please check NAMPT and NMNAT1-3.

Again, we appreciate this valuable concern and to accommodate the Reviewer´s concern we have performed several new experiments. First, we measured NAD+ levels in heart tissue after high dose NR, and there is indeed a tendency to an increase in NAD+ levels, however, not significant (Figure 5 – supplement 1). Next, we measured gene expression levels NAMPT and NMAT 1-3 with qPCR in heart tissue of mutUNG1 expressing mice and Wt controls, and no significant differences were found between the two genotypes with and without NR treatment.

Our findings therefore indicate that accumulation of NAM is not caused by reductions in the salvage pathway, but that it might, at least in Wt mice, be related to high sirtuin activity (Figure 5 – supplement 3A-D). However, we have also underscored that these important, but also complicated issues need further investigations.